# Analytical Cross Section Approximation for Electron Impact Ionization of Alkali and Other Metals, Inert Gases and Hydrogen Atoms

**Rusudan I. Golyatina [1] and Sergey A. Maiorov [1,2,*]**

[1] Prokhorov General Physics Institute of the Russian Academy of Sciences, 38, Vavilov Str., 119991 Moscow, Russia; rusudan@intemodino.com

[2] Joint Institute for High Temperatures of the Russian Academy of Sciences, 13, Izhorskaya Str., Bd. 2, 125412 Moscow, Russia

* Correspondence: mayorov_sa@mail.ru

**Abstract:** The paper presents an analysis of data on the cross sections of electron impact ionization of atoms of alkali metals, hydrogen, noble gases, some transition metals and Al, Fe, Ni, W, Au, Hg, U. For the selected sets of experimental and theoretical data, an optimal analytical formula is found and approximation coefficients are calculated. The obtained semi-empirical formula reproduces the values of the ionization cross sections in a wide range of energies with an accuracy of the order of error of the available theoretical and experimental data.

**Keywords:** electron atomic collisions; ionization cross section; approximation of cross sections; alkali metals; noble gases

## 1. Introduction

The values of electron-atomic collision cross sections are used in various applications of gas-discharge plasma. The bibliography on cross sections of electron-atomic collisions has thousands of works, and probably an exhaustive review and selection of data is contained in the works [1–6]. However, it should be borne in mind that a critical analysis of the results of experimental data in the review work is very difficult due to the fact that the errors given in the original works of the order of 1–3% differ from each other sometimes by 50%. Therefore, in the review work, only a comparative analysis of the results obtained is really possible, which shows that at best, the relative errors of measuring cross sections are of the order of 5–10%, and more often 20–50%, sometimes reaching 100%.

The most convenient form of presenting experimental and computational-theoretical data is the selection of analytical approximations for them. Analytical approximation is the most convenient and simple for computer modeling, for obtaining values at intermediate points. In addition, it allows you to analyze the accuracy of the asymptotic approximation. We started a critical analysis and evaluation of the cross sections for electron scattering by noble gas atoms in a wide energy range in [7–10], where we found approximations for the cross sections of elastic and inelastic collisions of electrons with rare gas atoms. From a large number of experimental and calculated data on ionization cross-sections, by comparative analysis, we selected the data for approximation by our analytical dependence. Ionization by electron impact from the ground state of the atom is, perhaps, the most frequent method for the formation and maintenance of a gas-discharge plasma. With a large excess of the electron energy above the ionization threshold, both experimental methods and theory provide good accuracy in measuring cross sections. However, there are practically no experimental data for low energies, and therefore, it is difficult to speak about the accuracy of theoretical calculations in this energy range.

As before, when choosing data for approximating ionization cross sections, we limited ourselves to considering ionization from the ground state, which is sufficient for modeling applied problems of gas-discharge plasma.

## 2. Approximation of the Ionization Cross Section

The formulation of the problem of finding an analytical approximation of the ionization cross section of an atom by an electron impact is based on the use of known analytical estimates, the results of experimental measurements and numerical quantum mechanical calculations. In 1912, Thomson proposed the dependence of the ionization cross-section on the electron energy of the following form [11]:

$$\sigma_{ionization}(\varepsilon) = \frac{\pi e^4}{\varepsilon}\left(\frac{1}{I} - \frac{1}{\varepsilon}\right) \equiv 4\pi a_0^2 \frac{Ry^2(\varepsilon - I)}{\varepsilon^2 I} \tag{1}$$

which is obtained for the case of a stationary valence electron at the energy of the incident electron $\varepsilon > I$. It gives a linear increase in the ionization cross section with a small excess of the collision energy over the ionization potential and reaches the maximum value $\sigma_{\max} = \pi e^4/4I^2$ at the energy of the incident electron $\varepsilon = 2I$. Here, $e$—elementary charge, $a_0$—Bohr radius, $Ry$—ionization energy of a hydrogen atom. A more precise expression for the ionization cross section, which takes into account the spherically symmetric motion of the valence electron in the Coulomb field of the atomic residue, has the form [12]:

$$\sigma_{ionization}(\varepsilon) = \frac{\pi e^4}{\varepsilon}\left(\frac{5}{3I} - \frac{1}{\varepsilon} - \frac{2I}{3\varepsilon^2}\right) \tag{2}$$

In this case, the maximum value $\sigma_{max} \approx \pi e^4/2I^2$ at the energy of the incident electron $\varepsilon = 1.85I$.

For the first time, a semi-empirical formula for approximating the initial section $I < \varepsilon < 2I$ of the dependence of the ionization cross section on the energy of the incident electron was proposed by Compton and Van Voorhees in 1925 [13] $\sigma_{ionization}(\varepsilon) = C_i(\varepsilon - I)$. Wannier proposed a power dependence with the exponent equal to 1.127 to approximate the initial section: $\sigma_{ionization}(\varepsilon) = C_i(\varepsilon - I)^{1.127}$, $\varepsilon > I$, which takes into account the interaction of the free and bound electrons [14].

Lotz in [15,16] analyzed the experimental and theoretical data available at that time and proposed a formula based on the Bethe–Born approximation, which has the form

$$\sigma_{ionization}(x) = [A\ln x + \sum_{k=1}^{N} B_k(\Delta x/x)^k]/xI^2, \quad x = \varepsilon/I, \ \Delta x = x - 1, \ x > 1 \tag{3}$$

Since the first ionization potential $I$ can serve as a natural scale of energy in the collision of electron with atom, it is therefore convenient to introduce the dimensionless energy: $x = \varepsilon/I$, $\Delta x = x - 1$, $x > 1$, $A$, $B_k$—fitting constants. The Lotz Formula (3) takes into account the universal dependence of the cross section on the ionization potential and is consistent with the asymptotic behavior of the Bethe formula $\sigma_{ionization}(\varepsilon) = (B + A\ln\varepsilon)/\varepsilon I$ [17].

There are also a number of other approaches to calculating ionization cross sections. For example, in [18], the paper presents semi-classical formula which allows the satisfactory evaluation of ionization cross-section for ionization of atoms. Their formula consists of the classical binary encounter approximation and the Born–Bethe approximation. This approach is applied to the rare gases, atomic nitrogen, and fluorine. Their approach leads to a better agreement with experimental results than the previous classical and semi-classical methods.

A theoretical binary dipole (BED) model which does not contain adjustable parameters is considered in [19]. There is also considered a simplified version, the so-called binary-encounter-Bethe (BEB) model. Both types of cross sections approximations have three basic components: the electron exchange term, the hard collision term, and the dipole interaction

term. The ratios between these components were determined by requiring the asymptotic total ionization cross section to agree with the asymptotic form given by the Bethe theory.

In [20], the cross sections are computed using a combination of spherical complex optical potential formalism and complex scattering potential method. The results obtained for thirteen elements are presented in the form of tabular values and are in good agreement with available measurements and theoretical data. However, it should be noted that this good agreement again has an error of the order of 10–30%, which corresponds to the scatter of values from different sources. In addition, in the model used, it is necessary to know the cross sections for elastic collisions, and to calculate the ionization cross sections, a relation is introduced between the ionization and excitation cross sections with three adjustable parameters for each type of atom. According to the authors of the work, the error in determining the maximum value of the ionization cross section and the position of this peak is of the order of 10%. We took their data for manganese, for which the approximation we obtained is in much better agreement with the dependence of the maximum of the ionization cross section on the polarizability and ionization potential given there.

In [21], the calculated cross sections are obtained for electron–atom scattering processes represented by a complex potential. For tungsten, ionization cross sections are discussed in the electron energy region from threshold up to 5000 eV against the available data from the Deutsch–Märk formalism [18] and a semi-empirical complex scattering potential. Papers [20,21] contain a rather detailed analysis of various approaches to calculating the cross sections for electron–atom collisions (elastic and inelastic), and data are also given on the most reliable (according to the authors) experimental data.

Since for the numerical simulation of many problems in plasma physics, the most convenient form of representing the dependence of the ionization cross sections on energy is the analytical dependence, then, we made an attempt to approximate the dependence of the ionization cross section on energy by the following new formula:

$$\sigma_{ionization}(\Delta x) = \frac{\alpha \Delta x}{(1 + \beta \Delta x)^{\gamma}} \tag{4}$$

where $\alpha$, $\beta$, $\gamma$—fitting constants. For $\alpha = 4\pi a_0^2 R_y^2 / I^2$, $\beta = 1$, $\gamma = 2$, it coincides with Thomson's Formula (1). Usually, when approximating by the Lotz Formula (3), 2–3 terms are used, whereas in our Formula (4) there is only 1 term with 3 fitting factors. In addition, our formula does not use a logarithmic dependence, and the power dependence makes it much more convenient to use both for theoretical analysis and for computer simulation.

To determine the coefficients $\alpha$, $\beta$, $\gamma$, the problem of minimizing the root-mean-square deviation of the cross sections from their experimental values was solved by the standard method of coordinate descent:

$$\Delta^2 = \frac{1}{N} \sum_{i=1}^{N} \left[ \frac{\sigma_{fit}(x_i) - \sigma_{\exp}(x_i)}{\sigma_{\exp}(x_i)} \right]^2 \tag{5}$$

where $\Delta$—standard deviation, $\sigma_{exp}(x_i)$—experimental values, $\sigma_{fit}(x_i)$—calculated values in points $x_i$: $i = 1, \ldots, N$. Minimizing the relative deviation $[\sigma_{fit}(x_i) - \sigma_{\exp}(x_i)]/\sigma_{\exp}(x_i)$ instead of minimizing the simple deviation $\sigma_{fit}(x_i) - \sigma_{\exp}(x_i)$ has the advantage of giving the correct statistical weight to cross sections at low and high impact electron energy. The tables show the value of the standard deviation in a percentage.

## 3. Results

The characteristics of atoms and experimental data, the error and parameters of the approximation of the ionization cross sections, as well as the general characteristics of the ionization cross sections for the found approximations are collected in twelve columns of Tables 1–4. The first column contains the name of the element and atom number, then the static dipole polarizability and ionization potential, which characterize the properties of the outer electron shell of atoms. In the fourth and fifth columns are the energy range

and the number of points of the experimental data used, then the standard deviation of the approximation and the values of the approximation coefficients of the ionization cross sections. In the tenth and eleventh columns are the position of the maximum cross-section and the maximum cross-section according to the approximating formula; in the twelfth is the constant of the linear approximation of the initial section $C_{ion} = \alpha/I$ obtained from Formula (4). The data in all the tables for $\alpha$, $\beta$, $\gamma$, $\varepsilon_m$ and $\sigma(\varepsilon_m)$ are received by us and are new.

**Table 1.** Characteristics of hydrogen atoms and alkali metals, error and parameters of approximation, general characteristics of cross sections according to the found approximations.

| Atom | | | Experiment | | | Approximation | | | Cross Section Value | | |
|---|---|---|---|---|---|---|---|---|---|---|---|
| No, Symbol | $K_0$, $a^3_0$ | $I$, eV | $\varepsilon_1 \div \varepsilon_N$, eV | $N$ | $\Delta$, % | $\alpha$, Å$^2$ | $\beta$ | $\gamma$ | $\varepsilon_m$, eV | $\sigma(\varepsilon_m)$, Å$^2$ | $C_i$, Å$^2$/eV |
| 1, H | 4.5 | 13.595 | 14.6 ÷ 3998 | 10 | 2% | 0.827 | 0.351 | 1.91 | 56.2 | 0.628 | 0.061 |
| 3, Li | 162 | 5.392 | 50 ÷ 500 | 6 | 1% | 5.72 | 0.500 | 1.67 | 21.5 | 3.71 | 1.06 |
| 11, Na | 162 | 5.139 | 6 ÷ 50 | 21 | 3% | 9.56 | 0.521 | 1.90 | 16.1 | 4.93 | 1.86 |
| 19, K | 287 | 4.339 | 50 ÷ 500 | 6 | 2% | 6.54 | 0.362 | 1.57 | 25.3 | 6.47 | 1.51 |
| 37, Rb | 310 | 4.176 | 50 ÷ 500 | 6 | 7% | 4.83 | 0.206 | 1.82 | 28.9 | 6.69 | 1.16 |
| 55, Cs | 385 | 3.893 | 50 ÷ 500 | 6 | 3% | 3.87 | 0.127 | 1.81 | 41.7 | 8.76 | 0.994 |

**Table 2.** Characteristics of noble gas atoms, error and parameters of approximation, general characteristics of cross sections according to the found approximations.

| Atom | | | Experiment | | | Approximation | | | Cross Section Value | | |
|---|---|---|---|---|---|---|---|---|---|---|---|
| No, Symbol | $K_0$, $a^3_0$ | $I$, eV | $\varepsilon_1 \div \varepsilon_N$, eV | $N$ | $\Delta$, % | $\alpha$, Å$^2$ | $\beta$ | $\gamma$ | $\varepsilon_m$, eV | $\sigma(\varepsilon_m)$, Å$^2$ | $C_i$, Å$^2$/eV |
| 2, He | 1.383 | 24.587 | 30 ÷ 4000 | 21 | 3% | 0.365 | 0.287 | 1.91 | 119 | 0.34 | 0.015 |
| 10, Ne | 2.68 | 21.564 | 30 ÷ 4000 | 21 | 6% | 0.373 | 0.136 | 2.00 | 180 | 0.68 | 0.017 |
| 18, Ar | 11.08 | 15.759 | 20 ÷ 4000 | 23 | 3% | 2.92 | 0.285 | 1.86 | 80 | 2.83 | 0.185 |
| 36, Kr | 16.74 | 13.996 | 20 ÷ 4000 | 22 | 3% | 3.51 | 0.269 | 1.80 | 79 | 3.80 | 0.251 |
| 54, Xe | 27.06 | 12.127 | 15 ÷ 4000 | 23 | 6% | 4.30 | 0.259 | 1.76 | 74 | 4.99 | 0.355 |

**Table 3.** Characteristics of atoms of transition metals, error and parameters of approximation, general characteristics of cross sections according to the found approximations.

| Atom | | | Experiment | | | Approximation | | | Cross Section Value | | |
|---|---|---|---|---|---|---|---|---|---|---|---|
| No, Symbol | $K_0$, $a^3_0$ | $I$, eV | $\varepsilon_1 \div \varepsilon_N$, eV | $N$ | $\Delta$, % | $\alpha$, Å$^2$ | $\beta$ | $\gamma$ | $\varepsilon_m$, eV | $\sigma(\varepsilon_m)$, Å$^2$ | $C_i$, Å$^2$/eV |
| 22, Ti | 148 | 6.83 | 10 ÷ 10,000 | 18 | 4% | 19.1 | 0.654 | 1.85 | 19.1 | 8.17 | 2.80 |
| 25, Mn | 101 | 7.432 | 8.0 ÷ 2000 | 29 | 8% | 8.39 | 0.413 | 1.62 | 36.4 | 6.9 | 1.13 |
| 26, Fe | 88 | 7.90 | 9.0 ÷ 200 | 59 | 5% | 14.8 | 1.15 | 1.44 | 23.5 | 5.3 | 1.87 |
| 28, Ni | 67 | 7.663 | 10 ÷ 10,000 | 17 | 6% | 6.04 | 0.405 | 1.86 | 29.7 | 4.12 | 0.787 |
| 29, Cu | 40 | 7.724 | 9.0 ÷ 200 | 59 | 2% | 6.86 | 0.645 | 1.52 | 31.0 | 4.0 | 0.891 |
| 46, Pd | - | 8.33 | 10 ÷ 10,000 | 17 | 3% | 3.09 | 0.146 | 1.89 | 72.3 | 5.7 | 0.371 |
| 47, Ag | 67 | 7.574 | 8.0 ÷ 200 | 60 | 8% | 7.65 | 0.565 | 1.46 | 36.7 | 5.45 | 1.01 |
| 74, W | 115 | 7.98 | 15 ÷ 5000 | 17 | 6% | 7.12 | 0.379 | 1.62 | 42.0 | 6.39 | 0.891 |
| 79, Au | - | 9.223 | 16 ÷ 21,800 | 15 | 8% | 16.5 | 0.265 | 1.86 | 49.7 | 17.2 | 1.79 |

**Table 4.** Characteristics of atoms of some metals, error and parameters of approximation, general characteristics of cross sections according to the found approximations.

| Atom | | | Experiment | | | Approximation | | | | Cross Section Value | | |
|---|---|---|---|---|---|---|---|---|---|---|---|---|
| No, Symbol | $K_0$, $a^3_0$ | $I$, eV | $\varepsilon_1 \div \varepsilon_N$, eV | $N$ | $\Delta$, % | $\alpha$, $\text{Å}^2$ | $\beta$ | $\gamma$ | $\varepsilon_m$, eV | $\sigma(\varepsilon_m)$, $\text{Å}^2$ | $C_i$, $\text{Å}^2/\text{eV}$ |
| 4, Be | 37.8 | 9.323 | 9.4 ÷ 112 | 28 | 13% | 3.22 | 0.338 | 2.20 | 32.3 | 2.1 | 0.346 |
| 12, Mg | 72 | 7.646 | 8.0 ÷ 200 | 60 | 3% | 13.7 | 0.714 | 1.87 | 19.9 | 5.3 | 1.79 |
| 13, Al | 162 | 5.986 | 6.0 ÷ 200 | 60 | 5% | 11.6 | 0.337 | 1.80 | 28.2 | 9.97 | 1.93 |
| 14, Si | 37 | 8.157 | 9.0 ÷ 200 | 59 | 4% | 9.97 | 0.503 | 1.61 | 34.8 | 6.82 | 1.22 |
| 80, Hg | 34.4 | 10.434 | 10.9 ÷ 29.2 | 36 | 20% | 1.00 | 0.222 | 1.74 | 73.9 | 1.37 | 0.096 |
| 82, Pb | - | 7.415 | 8.0 ÷ 200 | 60 | 7% | 12.8 | 0.592 | 1.52 | 31.5 | 8.20 | 1.74 |
| 92, U | - | 5.65 | 7.5 ÷ 500 | 30 | 19% | 5.04 | 0.329 | 1.73 | 29.2 | 4.72 | 0.89 |

Table 1 shows the results for the alkali metals and hydrogen atoms, because the hydrogen atom has one electron on the outer shell, as well as alkali atoms. The standard deviation of the found approximations is of the order of 2–7%, which corresponds in order of magnitude to the error of the initial data.

As a reference in Table 2 shows, similar data for noble gases were obtained in our previous work [22].

Table 3 shows the results for atoms of some transition metals. The experimental and theoretical data for Ti, Ni, and Pd were taken from [23]; Mn—from [20]; Fe, Cu, Ag—from [24]; W—from [21]; Au—from [25].

Table 4 shows the results for atoms of some metals, which are often used in various technological processes as working materials (for example, in the processes of etching or sputtering in microelectronics). Metal vapors often appear in the plasma as impurities due to sputtering of structural elements of installations (walls, cathodes, etc.). Experimental and theoretical data for beryllium are taken from [26]; Mg, Al, Si, Pb—from [24]; Hg—[27]; U—[28].

The results shown in Tables 1–4 allow for a critical analysis of both experimental and theoretical-computational data. Moreover, by interpolation or extrapolation, they can be used to obtain an estimate of the ionization cross sections for metal vapors for which data are not available. In particular, for platinum, experiments with which are carried out in a gyrotron discharge [29], the following values of the coefficients for approximating the cross sections can be recommended: $\alpha = 12$, $\beta = 0.32$, $\gamma = 1.72$.

## 4. Discussion

The experimental data and the approximating curves are shown in Figures 1–6 for H, Li, Na, K, Rb and Cs, respectively. In all plots, the experimental and theoretical values of the cross sections are shown by markers, and the solid curve is the found approximations. In addition, all figures show the values of the errors of the corresponding approximations. Solid curves in all the figures are original and obtained in this work.

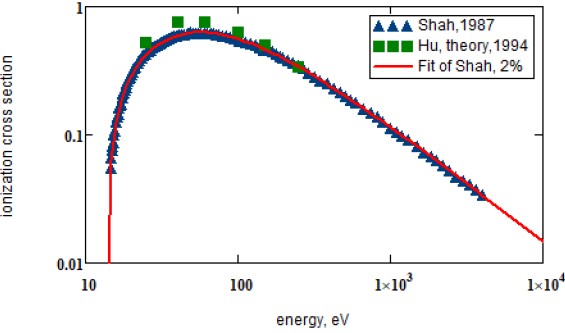

**Figure 1.** Electron impact ionization cross sections of hydrogen in $\text{Å}^2$.

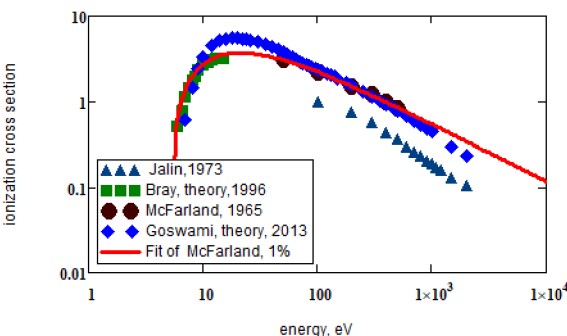

**Figure 2.** Electron impact ionization cross sections of lithium in Å$^2$.

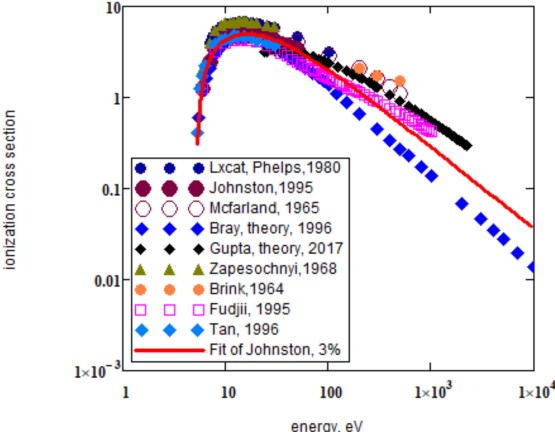

**Figure 3.** Electron impact ionization cross sections of sodium in Å$^2$.

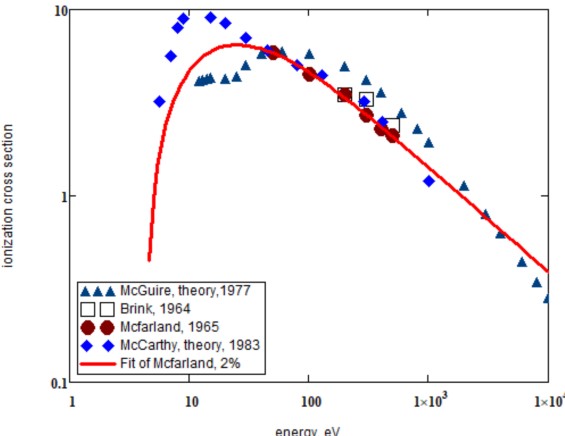

**Figure 4.** Electron impact ionization of potassium in Å$^2$.

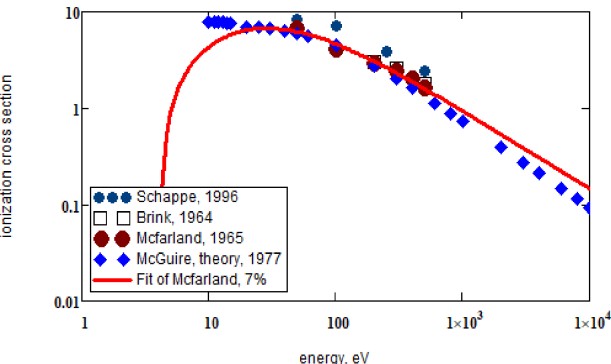

**Figure 5.** Electron impact ionization cross sections of rubidium in Å$^2$.

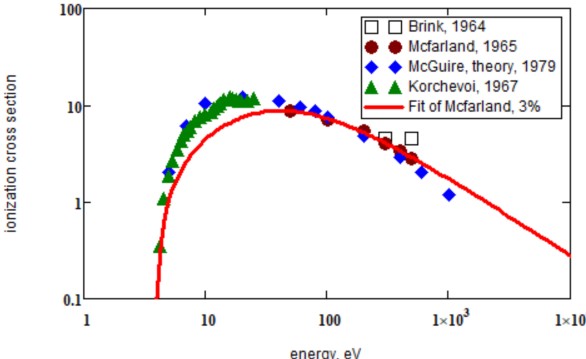

**Figure 6.** Electron impact ionization cross sections of cesium in Å$^2$.

*Hydrogen.* For hydrogen, there are many data, both theoretical and experimental, obtained with good accuracy. Figure 1 shows the data from the works [30,31]. The approximation coefficients for the values of the ionization cross sections from [30] are in Table 1.

*Lithium.* The data for lithium ionization cross sections are taken from [32–35] and are shown in Figure 2. The approximation is made for the values of the cross sections from [32]. In this work, there are only 6 experimental values, but the obtained approximation is in very good agreement with the data from [34] for low (<15 eV) energies. Therefore, we have chosen these data to determine the coefficients of analytical approximation of the ionization cross section.

*Sodium.* For sodium, there are many data [32,34,36–42]. They are shown in Figure 3. A large scatter is observed for energies above 100 eV. For the approximation, the ionization cross sections were taken from [38]. The obtained approximation is in good agreement with the majority of other authors even at $\varepsilon > 100$ eV.

*Potassium.* Data from the works [32,36,37,43] were analyzed for potassium; they are shown in Figure 4. The approximation is made for the values of the cross sections from [32].

*Rubidium.* The data for the ionization cross sections of rubidium were taken from the works [32,36,37,44]. The approximation is made for the data from [32]; see Figure 5.

*Cesium.* The data for the ionization cross sections of cesium were taken from the works [32,36,45,46]. The approximation is made for the data from [32]. This approximation is in good agreement with the theory [45] in the range 40 eV < $\varepsilon$ < 150 eV; see Figure 6.

Thus, in this work, based on a review and critical analysis of the available experimental and theoretical data on the cross sections of electron impact ionization of alkali metal and hydrogen atoms, we have suggested new analytical approximation formula that have an error of the same order of magnitude as the experimental data. As preliminary results, similar analytical approximations were obtained for the ionization cross sections of atoms of some transition metals, and for atoms of some other metals, which are often used in various technological processes as working materials.

**Author Contributions:** R.I.G. performed fitting calculations, S.A.M. performed modeling. All authors equally contributed to preparation of the manuscript. All authors have read and agreed to the published version of the manuscript.

**Funding:** This work was funded by the State Assignment GZ BV10-2021 "Study of Innovative Synthesis of Micro- and Nanoparticles with a Controllable Composition and Structure Based on Microwave Discharge in Gyrotron Radiation".

**Institutional Review Board Statement:** Not applicable.

**Informed Consent Statement:** Not applicable.

**Data Availability Statement:** Numerical data for cross sections can be obtained from the authors upon request.

**Conflicts of Interest:** The authors declare no conflict of interest.

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
