# Peer review of "Analytical Cross Section Approximation for Electron Impact Ionization of Alkali and Other Metals, Inert Gases and Hydrogen Atoms"

_atoms, doi:10.3390/atoms9040090_

Round 1
Reviewer 1 Report
I have several suggestions which I believe would improve this manuscript.
Because this paper presents results for hydrogen and noble gases, which are not metals, I'd suggest a change to the title.
On lines 29-31 the paper states that reviews are difficult because "...only the authors of the works can have the necessary initial data..." I think this needs clarification. As long as the data is available in the published works the review of it should be possible. What is the significance of "initial" data as opposed to any other cross section data?
The sentence on lines 36-37 states that analytical approximations is the "most convenient form" of presenting the data. I think this statement requires some justification.
On lines 41-42 the text refers to the "...scope of applicability of the selected analytical dependencies." What does this mean? Some clarification is needed.
On line 46 the text states that for low energies "...the accuracy of theoretical calculations is also low." I think this statement should be justified.
On line 54, a reference for the 1912 work of Thomson would be helpful.
On line 63, a reference for Eq. (2), or statement of where it comes from, would be helpful.
The paragraph in lines 65-69 make two references to "the initial section." A brief explanation of that term would be helpful.
On line 81 it states that for the listed parameters Eq. (4) coincides with Eq. (1). This is only true if Ry = 1. If such units are being employed in this work, it should be explicitly stated somewhere.
Regarding Eq. (5), there should be a statement that defines the quantity Δ along with the other quantities specified on line 85.
Concerning the determination of the parameters, there are many minimization schemes so I believe the text should describe how this particular work is carried out.
On line 93 I'd suggest changing "its number" to "its atomic number" and "the polarizability" to "the static dipole polarizability" or whichever polarizability is being used.
It appears that Table 1 and Figs. 1-6 go together. On lines 148-49 it states that the approximation coefficients for hydrogen are from another author. This was not clear much earlier in the text when Table 1 was being presented. Furthermore, the approximation in all of the figures were from other authors. That implies that none of the data in Table 1 is original to this work, not just for hydrogen. If that is the case it should also be made clear somewhere.
A related issue is that none of the figures appear to contain results original to this work. I believe the paper would be significantly better if the emphasis was more on original results than on presenting the work of others.
It is generally not very clear which results are new and which are from previous work or other investigators. It seems that the approximation coefficients in Tables 3 and 4 are new results, but the readers should not have to guess at this.
Reviewer 2 Report
The article presents an analytical fitting procedure for the selected set of atoms using existing theoretical/experimental data. I am not sure about the usefulness of the methodology as it simply duplicates an existing data. It cannot predict cross sections for atoms where data is not available.
The paper gives good analytical fit of ionization cross section values, but the questions are,
- "how this analytical form can help to visualize the scattering phenomena.
- One can understand formula and
their terms, but where is the physical concept of scattering? This is needed to predict cross sections. - What are the reasons to choose this analytical form over the previous ones?
- Is there any need for modification to previous analytical form? Or is the present analytical form is just a way to calculate the values which shows good relation to previous outcomes?
In conclusion, I have strong reservation in recommending the article for publication in this journal.
Reviewer 3 Report
Thank you very much for this interesting read.
You present an interesting new approach to fitting sparse experimental data for use with a wider range of energies. Especially in cases where data is available for only 6 different energies, your approach is very valuable. Also your fitting procedure does not rely on other hard-to-measure quantities, but only on known data for ionization potential and polarizability.
Before I can recommend publication, I would like you to address two minor issues. The first (and smaller one) is that figures 2,4 and 6 have cut-off figure captions, missing the type of atom. Also the labels on the figure axis are too small to read.
The second issue is the way that you derive the error of your fit. First, I think the term "uncertainty" might be a better choice, as there is no "error" in your mathematical formula. You minimize the error of the fit for each individual pair of values, but the result is better described as the uncertainty, I think. But I could be wrong here.
Secondly, I would like to ask you to add a sentence or two (not more) to briefly recapitulate, how you calculate this error. It was not clear to me from reading the manuscript how your method relates to relative errors and why the weight of the contributions at different energy ranges differ.
Apart from these two issues, and a bit of language editing, I think the paper presents a very valuable addition to the scientific toolbox for not only the plasma-community but also other related fields that have to make do with very limited data on electron-impact cross sections.
Author Response
Response to Reviewer 3 Comments
Point 1: The first (and smaller one) is that figures 2,4 and 6 have cut-off figure captions, missing the type of atom. Also the labels on the figure axis are too small to read.
Response 1: Fixed
Point 2: The second issue is the way that you derive the error of your fit. First, I think the term "uncertainty" might be a better choice, as there is no "error" in your mathematical formula. You minimize the error of the fit for each individual pair of values, but the result is better described as the uncertainty, I think. But I could be wrong here.
Secondly, I would like to ask you to add a sentence or two (not more) to briefly recapitulate, how you calculate this error. It was not clear to me from reading the manuscript how your method relates to relative errors and why the weight of the contributions at different energy ranges differ.
Apart from these two issues, and a bit of language editing, I think the paper presents a very valuable addition to the scientific toolbox for not only the plasma-community but also other related fields that have to make do with very limited data on electron-impact cross sections.
Response 2:
References 3 and 6 describe The LXCat Project , so we replaced the reference 6 to
Brusa R.S.; Karwasz, G.P.; and Zecca, A. Analytical partitioning of total cross sections for scattaring on noble gases. . Phys. D 1996, 38, 279287. DOI: 10.1007/s004600050092.
In this paper, the problem of selecting analytical approximations for cross sections of collisions of an electron with an atom is considered. The method described in it for determining the coefficients in the approximating formula completely coincides with ours. And we have slightly changed the wording describing the fit error that follows after formula (6).
It was
where Δ - root-mean-square deviation, ϭexp(xi) experimental values, ϭfit(xi) calculated values in points xi: i=1,…, N. Minimizing the relative error instead of minimizing the absolute error allows us to more correctly take into account the statistical weight of the cross sections at low and high electron energies.
Become
where Δ – standard deviation, ϭexp(xi) are the experimental values and ϭfit(xi) are calculated cross sections in points xi: i=1,…, N. Minimizing the relative deviation instead of minimizing the simple deviation has the advantage of giving the correct statistical weight to cross sections at low and high impact electron energy. The tables show the value of the standard deviation as a percentage, i.e. Δ 100%.

Round 2
Reviewer 2 Report
The reply to my earlier comments are as expected, and not convincing enough to recommend for publication.
